# Exploring the Moderating Effect of Positive and Negative Word-of-Mouth on the Relationship between Health Belief Model and the Willingness to Receive COVID-19 Vaccine

**DOI:** 10.3390/vaccines11061027

**Published:** 2023-05-26

**Authors:** Shih-Wei Wu, Pei-Yun Chiang

**Affiliations:** Department of Business Management, National Taipei University of Technology, Taipei 10608, Taiwan; swu@ntut.edu.tw

**Keywords:** Health Belief Model, positive and negative word-of-mouth, vaccine receive willingness

## Abstract

This study indicates that the positive and negative effects of word-of-mouth (WOM) have an impact and moderating effect on vaccine uptake willingness, which is important to exploring the factors that affect vaccine uptake. We further analyzed the differences in the impact relationship between variables through questionnaire research. Based on the Health Belief Model (HBM) widely used to explore global health issues, this study focuses on Taiwanese residents and uses a questionnaire survey method. Furthermore, this study investigates the impact of various factors in the HBM on the willingness to receive the COVID-19 vaccine when faced with positive and negative word-of-mouth evaluations from the perspective of vaccine recipients, as well as whether WOM evaluations have an interference effect, along with the differences between variables. Practical recommendations are provided based on the research results, which can serve as a reference for future vaccine promotion programs and health promotion. By improving the national vaccination rate and achieving herd immunity, we aim to increase the persuasiveness of word-of-mouth on public healthcare decision-making. We also hope to provide a basis for health promotion and encourage people to make informed decisions about vaccination.

## 1. Introduction

Since 2019, the COVID-19 virus has rapidly spread from China to the rest of the world, becoming one of the most widespread pandemics in modern human history. In a short period, the global number of confirmed cases and fatalities grew, making it a leading cause of death. To substantially reduce the risk of severe illness and death, countries relied on vaccine development and the production of oral medications. Vaccination can help alleviate the burden of the disease, reduce pressure on healthcare systems, and prevent outbreaks. Moreover, when a large proportion of a population is immunized, disease transmission is significantly decreased and herd immunity can be achieved [1,2].

Vaccination acts as a protective measure that builds individual’s defenses and disrupts transmission chains. It bolsters the body’s immune system and is the most effective and cost-efficient method to control infectious diseases. Vaccines can successfully prevent severe symptoms after infection, thus reducing hospitalization and death rates [2,3,4]. Numerous past empirical studies have used the Health Belief Model to explore changes in health behaviors during the SARS and H1N1 outbreaks. These studies show that when people are convinced of a pandemic’s severity, they perceive an increased risk of illness, trust the effectiveness of preventive measures, and consider the costs of adopting such behaviors as low, making them more likely to engage in preventive actions [5].

A review of the literature on vaccine acceptance reveals that the “Health Belief Model” has been widely discussed as a theoretical basis for preventive health behaviors since the 1970s. Many researchers have used this theory to study vaccine acceptance in relation to H1N1, measles, human papillomavirus, and rabies vaccines [6,7,8]. Wong [9] applied the Health Belief Model to investigate the acceptance of COVID-19 vaccines among the Hong Kong population, while Ao [10] employed the model to examine the acceptance of COVID-19 vaccines among adults in Lilongwe, Malawi’s capital.

Earlier studies have confirmed that online WOM and recommendations in healthcare impact patients’ medical decisions and the adoption of new medications [11,12]. The research findings mentioned above indicate that prior studies using the Health Belief Model for COVID-19 vaccine-related research have primarily focused on specific populations in certain areas. However, a significant gap exists in the literature regarding the relationship between the Health Belief Model, vaccine acceptance, and positive and negative WOM evaluations.

Although medical word-of-mouth is a common phenomenon, previous studies in both the medical field and vaccine acceptance have rarely addressed the influence of WOM evaluations. Additionally, research has explored patient satisfaction and the effects of service quality, WOM, and trust on satisfaction with medical services [13]. In summary, given the variations in positive and negative WOM evaluations, public perceptions of vaccination may be affected. This study aims to address the knowledge gap in the literature by using positive and negative types of WOM as moderating variables to investigate how positive and negative WOM evaluations may influence people’s health beliefs and vaccine acceptance amid the ongoing COVID-19 pandemic. The goal is to further analyze the differences in the relationships among the various factors.

## 2. Literature Review

### 2.1. Health Belief Model, HBM

The Health Belief Model is frequently employed to examine care and medication-taking behaviors among elderly individuals with chronic illnesses, as well as to conduct cross-sectional research on behavioral influences and predictions. It is also utilized in health prevention programs and vaccination research studies. The results indicate that the Health Belief Model can effectively predict and analyze health behaviors.

Perceived susceptibility refers to an individual’s subjective evaluation of the likelihood of getting a disease or confidence in a diagnosis. A stronger perception of susceptibility is linked to a greater health belief and an increased likelihood of engaging in health prevention behaviors. This perception varies among individuals due to differing opinions, resulting in significant differences in vulnerability to a disease. A study by Delshad [14] found that all aspects of the Health Belief Model were associated with epidemic prevention behaviors and predicted these behaviors significantly.

Perceived seriousness refers to an individual’s assessment of the severity of a disease, including the potential harm to their health and social well-being. People’s perceptions of the severity of a disease can differ considerably, and those who underestimate the risk may engage in unhealthy behaviors. This evaluation includes beliefs about the disease and its impact on work and social roles relevant to the individual. Perceived seriousness, along with perceived susceptibility, are cognitive variables that can be influenced by education and knowledge.

Perceived benefits of taking action refer to an individual’s initial assessment of whether a particular action can reduce their susceptibility and severity to a disease while preserving personal health benefits. This belief is influenced by social norms and pressures, which can lead to different actions being embraced. However, the perceived benefits of action may sometimes be influenced by group norms and pressures, as noted by Adams [15].

Perceived barriers to taking action refers to an individual’s assessment of the challenges or obstacles they may face when undertaking a specific action, such as inconvenient transportation, high costs, physical or mental discomfort, unsafe side effects, and negative media coverage. Conflicting information and frequent updates on the COVID-19 situation were perceived as barriers by 65.9% of respondents, according to a study by Jose [16]. Shahnazi [17] highlighted that perceived barriers and self-efficacy were the most significant factors influencing COVID-19 prevention behaviors, with a strong correlation between the two.

Cues to action refer to the motivation or triggers that prompt individuals to take action, which can be internal, such as physical discomfort or symptoms, or external, such as advice from friends and family, social media, healthcare professionals, and health education campaigns. The intensity of the stimulus needed to initiate action depends on the situation. Several studies have investigated factors influencing the intent or behavior of getting vaccinated, including information from healthcare providers and health education, media advocacy, frequent exposure to related information, friends and family suffering from the disease, and individuals with middle to upper socioeconomic status, who are more likely to receive more cues and pay more attention to relevant information.

Self-efficacy refers to an individual’s belief in their ability to perform a specific health behavior in different situations and is part of the Health Belief Model. The stronger a person’s self-efficacy for a particular task, the more confident they are in their ability to complete the task. This confidence leads to a greater willingness to continue participating and putting in more effort to achieve the desired outcome. Many studies have found a significant correlation between self-efficacy and epidemic prevention behaviors, meaning that the higher an individual’s confidence in adopting prevention measures, the more likely they are to engage in such behaviors. Researchers such as Barakat [18], Fathian-Dastgerdi [19], Shahnazi [17], and Koesnoe [20] have demonstrated that higher self-efficacy and perceived benefits of action lead to a greater willingness to engage in epidemic prevention measures.

According to the motivations and literature mentioned earlier, this study derives the following research hypotheses:

**H1.** 
*There will be a significant impact of health beliefs (Perceived susceptibility, Perceived seriousness, Perceived benefits to taking action, Perceived barriers to taking action, Cue to action, and Self-efficacy) on the willingness to receive the COVID-19 vaccine.*


### 2.2. Positive and Negative Word-of-Mouth

The concept of “word-of-mouth” dates back to the 1950s, when it was first discovered and began to be studied academically. Word-of-mouth is an important form of communication that involves the flow of information through interpersonal sources or orally transmitted information. Initially, word-of-mouth refers to the exchange of information through spoken communication between people who are discussing a brand, product, or service without any intention of promoting it commercially. It refers to everyday interactions between people, generating a non-commercial form of communication through natural exchange, and discussing various pieces of information about brands, products, and services. It is also a way for people to understand specific brands, products, and personal ideas through sharing [21,22]. Under high-pressure care situations, consumers will seek information to reduce risk when using medical services. Chang [23] also mentioned that consumers search for product- or service-related information online or offline before making a decision to reduce potential risks [24]. Erkan and Evans [25] found that the adoption of word-of-mouth positively correlates with purchase intention, meaning that as consumers’ willingness to adopt word-of-mouth increases, purchase intention also rises.

Word-of-mouth can be positive or negative and has a significant impact on consumer behavior, affecting purchase decisions and product sales. Positive word-of-mouth is associated with recommendation and persuasion, while negative word-of-mouth is linked to customer complaints and can harm a company’s reputation. Negative word-of-mouth can be damaging as it spreads quickly and may result in decreased consumption and brand attitudes. Therefore, companies should monitor and manage word-of-mouth to ensure positive perceptions of their products or services [26,27].

Based on the aforementioned literature, scholars do not have a consistent conclusion regarding whether positive or negative word-of-mouth has a greater impact, which makes this study even more valuable. According to the motivations and literature mentioned earlier, this study derives the following research hypotheses:

**H2.** 
*Positive word-of-mouth evaluations will have a significant moderating effect on the relationship between health beliefs (Perceived susceptibility, Perceived seriousness, Perceived benefits of taking action, Perceived barriers of taking action, Cue to action, and Self-efficacy) and the willingness to receive the COVID-19 vaccine.*


**H3.** 
*Negative word-of-mouth evaluations will have a significant moderating effect on the relationship between health beliefs (Perceived susceptibility, Perceived seriousness, Perceived benefits of taking action, Perceived barriers of taking action, Cue to action, and Self-efficacy) and the willingness to receive the COVID-19 vaccine.*


## 3. Methods

The purpose of this study is to investigate the impact of various variables of the Health Belief Model on the willingness to receive COVID-19 vaccinations, and to examine the effects of positive and negative word-of-mouth as intervening variables. Empirical research is conducted with selected participants who reside in Taiwan and have received COVID-19 vaccinations. The paper concludes with a description of the questionnaire distribution, sample characteristics, and methods of questionnaire content analysis. The conceptual framework is shown in Figure 1:

The nine research variables in this study are measured using the Likert 7-point scale. The questionnaire items are referenced from relevant literature and appropriate scales were selected for each, with the principle of not losing the original intention of the questions, and then making minor adjustments and modifications. SUS references the Walker [28] and Khalafalla [29] studies with four items. SER references the Walker [28] and Jones [30] studies with five items. BEN references the Walker [28] and Berni [31] studies with four items. BAR references the Walker [28] and Berni [31] studies with four items. CUE references the Walker [28] and Khalafalla [29] studies with five items. SEFF references the Yoo [32] and Koesnoe [20] studies with four items. PWOM and NWOM reference the Maisam [33], Popp [34], and An [35] studies with eight items. WRV references the Nga [36] and Ning [37] studies with a total of five items.

In this study, judgmental sampling was used to gather samples. Google online surveys were employed, and six private educational institutions assisted in distributing paper surveys to students and their parents for completion. Further discussions were held regarding survey distribution and collection methods. The survey distribution period lasted from 15 March 2023 to 25 March 2023. Out of the 500 surveys distributed, 433 valid responses were collected after excluding missing responses, blanks, incomplete answers, and invalid multiple selections, resulting in an effective response rate of 86.6%. After collecting the surveys, the study first carried out encoding and data archiving of the questionnaires to facilitate subsequent statistical analysis. SPSS 23.0 statistical software was chosen for data analysis and processing.

## 4. Results

### 4.1. Descriptive Statistics Analysis

A descriptive statistical analysis was conducted on various aspects of the questionnaire items, as shown in Table 1, to understand the background information of the research sample. The primary demographic variables analyzed in this section can serve as a foundation for subsequent statistical analyses.

### 4.2. Factor Analysis and Reliability Analysis

This study employs factor analysis and reliability analysis to evaluate the relevance and consistency of each dimension and item. To enhance rigor, the commonality of each item is extracted if it exceeds 0.5, making it appropriate for analysis. The common factors are obtained using the maximum variance method. When the overall dimensional item Cronbach’s alpha value surpasses 0.7, it indicates consistency, as displayed in Table 2.

### 4.3. Pearson Correlation Coefficient

In this study, Pearson correlation analysis was used to determine the correlation levels and significance among the variables. The internal consistency of the construct was assessed using the Composite Reliability (CR), with all CR values exceeding 0.8, indicating acceptable convergent validity. The Average Variance Extracted (AVE) was also found to be greater than 0.5, suggesting convergent validity between the latent variables and their respective measurement items. Discriminant validity was evaluated by comparing the square root of AVE to the correlation coefficients between dimensions, which ranged from 0.74 to 0.93, indicating good discriminant validity, as shown in Table 3.

### 4.4. Variance Analysis

#### 4.4.1. Independent Sample *t*-Test of Gender

Using an independent sample *t*-test, the study analyzed gender and each dimension. The study found that there were significant differences between gender and self-awareness of severity (*p* = 0.004), self-awareness of action benefit (*p* = 0.043), action cues (*p* = 0.003), and self-efficacy (*p* = 0.006) in these four parts. As well, all the average values of males are greater than those of females, as shown in Table 4.

#### 4.4.2. ANOVA of Age

The results of the analysis, as shown in the Table 5, indicate that age has a significant difference in the self-efficacy variable (F = 2.783, *p* < 0.05). However, there is no difference in the other variables due to different age groups. After post-hoc multiple comparisons, the Scheffe method was used for analysis and it was found that self-efficacy was significantly higher in those under 20 years old than in those between 31 and 40 years old.

#### 4.4.3. ANOVA of Education

According to the analysis results shown in Table 6, there are significant differences in the two variables, action cues (F = 3.059, *p* < 0.05) and self-efficacy (F = 4.528, *p* < 0.05), based on the level of education.

Other variables do not vary with differences in education levels. Post-hoc multiple comparisons were conducted using the Scheffe method, revealing that the behavioral cues for individuals with junior high school education are significantly higher than those with high school education. In addition, self-efficacy among individuals with a junior high school education is significantly higher than those with a high school education, and self-efficacy among individuals with a junior high school education is also significantly higher than those with a college education.

#### 4.4.4. ANOVA of Occupation

The results of the analysis, as shown in Table 7, indicate that occupation has a significant difference in the two variables of perceived mobility barriers (F = 2.003, *p* < 0.05) and PWOM (F = 2.139, *p* < 0.05). Other variables do not differ due to differences in occupation. The post-hoc multiple comparisons were analyzed using the Scheffe method and it was found that although there was a significant difference, the difference was extremely small, making it impossible to determine the difference.

### 4.5. Regression Analysis

The relationship between health belief variables and vaccine willingness was analyzed using linear regression, and the results are summarized in Table 8. It shows that perceived susceptibility, perceived severity, perceived benefits of action, perceived barriers to action, cues to action, self-efficacy, and both positive and negative word-of-mouth have a positive impact on COVID-19 vaccination intentions. Therefore, H1 is supported. Additionally, to examine the issue of multicollinearity among the variables, the Tolerance values for each variable are between 0 and 1, and the Variance Inflation Factor (VIF) is less than 2.2, indicating that there is no multicollinearity problem in this study.

### 4.6. Hierarchical Regression Analysis

This study employs hierarchical regression analysis to examine the impact of PWOM and NWOM on the willingness to receive COVID-19 vaccinations, in relation to six independent variables in the HBM. Furthermore, to avoid multicollinearity issues caused by correlations among the main variables, interaction terms need to be calculated by separately standardizing the independent variables and the moderator variables, and then multiplying them to examine the moderating effects. The results of the SPSS analysis are presented in Table 9, Table 10, Table 11, Table 12, Table 13, Table 14, Table 15, Table 16, Table 17, Table 18, Table 19 and Table 20.

Based on the research method of Aiken & West (1991), this study categorizes PWOM into high and low groups and illustrates the moderating effects, as shown in Figure 2 below.

Based on the research method of Aiken & West (1991), this study categorizes PWOM into high and low groups and illustrates the moderating effects, as shown in Figure 3 below.

## 5. Conclusions

Previous research on the influence of various independent variables on willingness in health behavior patterns has seldom considered the impact of word-of-mouth reviews. This study reveals that both positive and negative word-of-mouth can have an interference effect, even though some statistical results do not fully support the research hypotheses. Nonetheless, our findings offer an opportunity to investigate topics not covered in past research, aiming to fill the gap in the literature on word-of-mouth. Prior studies have indicated that the influence of positive and negative word-of-mouth information varies, leading to different conclusions. This research discovers that both positive and negative word-of-mouth information have significant interference effects in the dimension of behavioral barriers, with negative word-of-mouth having a more pronounced impact than positive word-of-mouth. This finding aligns with the majority of past research on word-of-mouth effects, which has shown that negative word-of-mouth has a greater influence than positive word-of-mouth in shaping consumer product attitudes or evaluating.

In the ANOVA statistical analysis of this study, it was found that the variables of marital status, residency status, and average monthly income had no significant effect. This study identified significant gender differences in perceived severity, perceived benefits of action, cues to action, and self-efficacy, with males scoring higher averages than females in all four aspects. Age differences revealed significant disparities in self-efficacy, as those aged 20 or younger had considerably higher self-efficacy than those aged 31 to 40. Education level displayed significant differences in cues to action and self-efficacy, with individuals educated up to junior high school having notably higher cues to action than those educated at the high school (vocational) level and higher than those educated at the university (professional) level. Additionally, the study explored the moderating effects of positive and negative word-of-mouth. As shown in Figure 2, under conditions of high positive word-of-mouth, action benefits have a negative impact on the willingness to receive the COVID-19 vaccine; in contrast, when positive word-of-mouth is lower, action benefits have a positive impact on the willingness to receive the COVID-19 vaccine. On the other hand, as can be seen from Figure 3 and Figure 4, regardless of whether positive or negative word-of-mouth is high or low, mobility barriers have a positive impact on the willingness to inject COVID-19 vaccines; in addition, situations with high negative word-of-mouth have a more severe impact.

Compared to the results of previous studies, Tadesse [38] found that factors most closely associated with employees’ preventive behaviors were monthly income, perceived behavioral barriers, cues to action, and self-efficacy. They also discovered that individuals with lower levels of cues to action and self-efficacy were less likely to adopt preventive behaviors. However, some studies have produced different results. For example, Shahnazi [17] found that perceived severity was not significantly related to preventive behavior, while perceived benefits, self-efficacy, behavioral barriers, and cues to action had significant effects. In contrast, Barakat [18] reported that cues to action were not significantly related to preventive behavior. Additionally, Fathian-Dastgerdi [19] found significant negative correlations between adolescents’ perceived susceptibility, behavioral barriers, and preventive behaviors. Shahnazi [17] discovered that while participants had higher perceived susceptibility, severity, benefits, and self-efficacy, their overall preventive behavior was ideal; however, perceived susceptibility and severity were not significantly related to preventive behavior. Furthermore, Alagili [39] found that only perceived susceptibility and severity were not significantly related to preventive behavior.

This study’s practical contributions include investigating the interference effects of positive and negative word-of-mouth reviews, and allowing future epidemic prevention efforts to better understand vaccine recipients’ psychological state after receiving such information. To improve overall epidemic prevention effectiveness, future messages can be promoted through various social media platforms, using vivid descriptions, increasing interactions with consumers, and presenting information through images and videos. Word-of-mouth effects can save costs associated with large-scale vaccine promotion campaigns and help increase vaccination rates more quickly.

However, this study has limitations, such as being cross-sectional, using an online structured questionnaire, and having a limited sample size. Future research should expand the geographical scope, increase the sample size, and consider other variables, such as personal characteristics and organizational culture. Long-term follow-up studies can analyze the public’s changes in vaccine-related knowledge, attitudes, and behavior as the epidemic evolves. Lastly, future research can focus on the influence of social media word-of-mouth on vaccination intentions.

## Figures and Tables

**Figure 1 vaccines-11-01027-f001:**
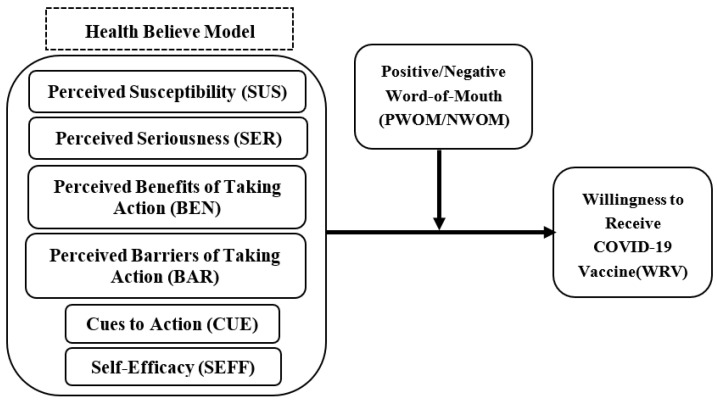
Conceptual framework for the hypothesized predictors of the willingness to receive a COVID-19 vaccine.

**Figure 2 vaccines-11-01027-f002:**
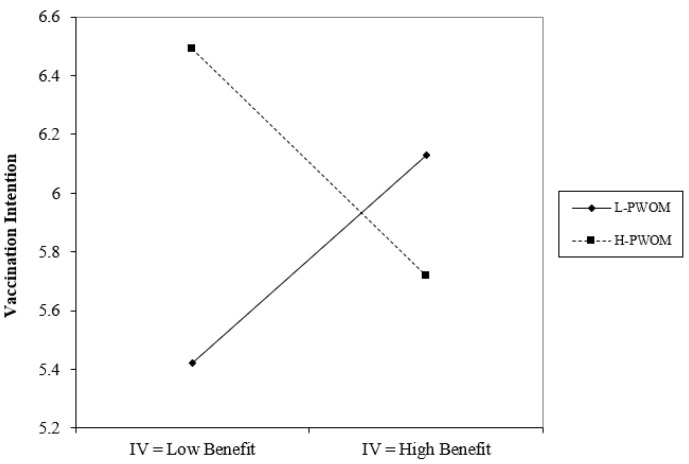
The interference effect of PWOM on the relationship between perceived benefits and willingness to receive COVID-19 vaccine.

**Figure 3 vaccines-11-01027-f003:**
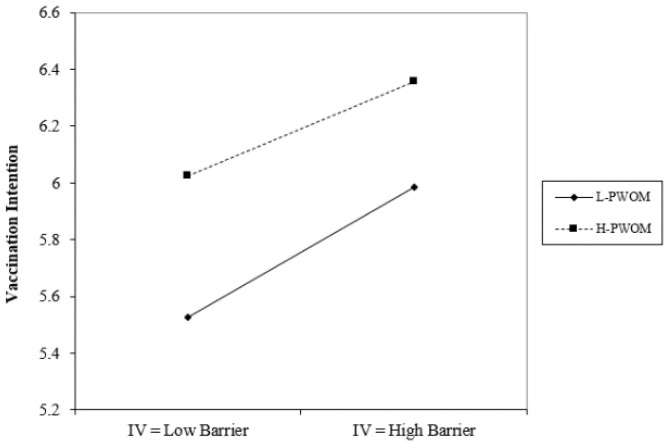
The interference effect of PWOM on the relationship between behavioral benefits and willingness to receive vaccine.

**Figure 4 vaccines-11-01027-f004:**
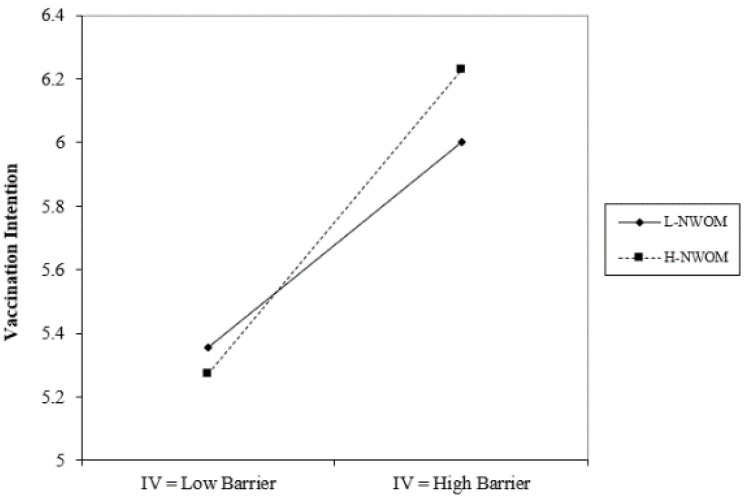
The interference effect of NWOM on the relationship between perceived barriers to action and willingness to receive vaccine.

**Table 1 vaccines-11-01027-t001:** Demographic characteristics of the respondents (*n* = 433).

Variables	Characteristics	Frequency	Percentage
COVID-19 Vaccinations	one vaccine dose	19	4.4
	two vaccine doses	87	20.1
	three vaccine doses or more	310	71.6
	Never	17	3.9
Confirmed COVID-19	Yes	240	55.4
	No	193	44.6
Gender	Male	196	45.3
	Female	237	54.7
Age	<20	16	3.7
	21–30	97	22.4
	31–40	130	30.0
	41–50	108	24.9
	51–60	39	9.0
	>61	43	9.9
Marital Status	Married	251	58.0
	Single	145	33.5
	Divorced	31	7.2
	Widowed	6	1.4
Education	Middle school and below	16	3.7
	High school	85	19.6
	Associate or bachelor	231	53.3
	Master and above	101	23.3
Current Living Status	Living alone	52	12.0%
	Live with spouse	86	19.9%
	Live with parents or children	275	63.5%
	Others	7	4.6%
Occupation	Students	23	5.3%
	Civil servant	33	7.6%
	Business	68	15.7%
	Workers	25	5.8%
	Manufacturing	76	17.6%
	Agriculture, forestry, fishery and animal husbandry	7	1.6%
	Service industry	119	27.5%
	Freelancer	44	10.2%
	Others	38	8.8%
Average Monthly Income (TWD)	<20,000	31	7.2
	20,001–35,000	65	15.0
	35,001–50,000	184	42.5
	50,001–65,000	88	20.3
	65,001–85,000	25	5.8
	85,001–100,000	17	3.9
	>100,000	23	5.3

**Table 2 vaccines-11-01027-t002:** Factor Analysis and Reliability Analysis of Each Variable.

Factors	Items	Factor Loadings	Communality	Explained Variation (%)	Cronbach’s α
SUS	1	0.642	0.822	82.72	0.724
	2	0.755	0.768		
	3	0.819	0.854		
	4	0.778	0.864		
SER	1	0.708	0.501	57.08	0.810
	2	0.804	0.646		
	3	0.782	0.612		
	4	0.748	0.560		
	5	0.732	0.536		
BEN	1	0.920	0.846	81.08	0.883
	2	0.932	0.868		
	3	0.848	0.719		
BAR	1	0.820	0.673	72.33	0.872
	2	0.868	0.754		
	3	0.863	0.745		
	4	0.849	0.721		
CUE	1	0.743	0.553	60.41	0.835
	2	0.761	0.580		
	3	0.830	0.688		
	4	0.826	0.682		
	5	0.720	0.518		
SEFF	1	0.781	0.610	55.70	0.730
	2	0.797	0.636		
	3	0.707	0.500		
	4	0.695	0.483		
WRV	1	0.770	0.594	70.92	0.861
	2	0.891	0.793		
	3	0.868	0.754		
	4	0.834	0.696		
PWOM	1	0.816	0.666	71.07	0.862
	2	0.890	0.791		
	3	0.880	0.774		
	4	0.782	0.611		
NWOM	1	0.929	0.864	86.20	0.964
	2	0.921	0.848		
	3	0.925	0.856		
	4	0.938	0.880		

**Table 3 vaccines-11-01027-t003:** Pearson Correlation Analysis of Each Variable.

	SUS	SER	BEN	BAR	CUE	SEFF	WRV	PWOM	NWOM	CR	AVE
SUS	0.75	0.615 **	0.133 **	0.379 **	0.274 **	0.319 **	0.191 **	0.250 **	0.194 **	0.84	0.56
SER		0.75	0.223 **	0.409 **	0.501 **	0.502 **	0.271 **	0.375 **	0.214 **	0.87	0.57
BEN			0.90	0.387 **	0.365 **	0.244 **	0.626 **	0.401 **	0.149 **	0.93	0.81
BAR				0.85	0.219 **	0.217 **	0.352 **	0.158 **	0.435 **	0.91	0.72
CUE					0.77	0.657 **	0.494 **	0.561 **	0.227 **	0.88	0.60
SEFF						0.74	0.436 **	0.472 **	0.284 **	0.83	0.55
WRV							0.84	0.590 **	0.175 **	0.91	0.71
PWOM								0.84	0.243 **	0.91	0.71
NWOM									0.93	0.96	0.86

** The correlation is significant at the 0.01 level (two-tailed). The diagonal values are the square root of the AVE values, and the upper triangle is the Pearson correlation coefficient.

**Table 4 vaccines-11-01027-t004:** Independent Sample *t*-test of Gender Variable.

Variables	Gender	Mean	Standard Deviation	*t*	Sig.
SUS	Male	5.564	0.888	−0.334	0.738
	Female	5.590	0.723		
SER	Male	5.779	0.784	2.876	0.004 *
	Female	5.567	0.743		
BEN	Male	5.915	0.862	2.026	0.043 *
	Female	5.740	0.922		
BAR	Male	5.478	1.062	−0.339	0.734
	Female	5.511	0.879		
CUE	Male	5.694	0.720	2.991	0.003 *
	Female	5.490	0.698		
SEFF	Male	5.762	0.685	2.754	0.006 **
	Female	5.580	0.679		
WRV	Male	5.768	0.805	0.530	0.596
	Female	5.727	0.800		
PWOM	Male	5.792	0.854	1.964	0.050
	Female	5.639	0.744		
NWOM	Male	5.2194	1.20030	−0.155	0.877
	Female	5.2363	1.06294		

* *p* < 0.05; ** *p* < 0.01.

**Table 5 vaccines-11-01027-t005:** One-Way ANOVA Table of Age Variable.

Variables	Age	Mean	Standard Deviation	F	*p*
SUS	<20	5.69	1.07	0.831	0.528
	21–30	5.64	0.84		
	31–40	5.51	0.73		
	41–50	5.59	0.84		
	51–60	5.42	0.77		
	>61	5.70	0.73		
SER	<20	6.06	0.80	1.389	0.227
	21–30	5.67	0.77		
	31–40	5.64	0.73		
	41–50	5.64	0.83		
	51–60	5.49	0.71		
	>61	5.74	0.72		
BEN	<20	6.15	0.76	1.603	0.158
	21–30	5.75	1.00		
	31–40	5.81	0.84		
	41–50	5.75	0.89		
	51–60	5.78	0.97		
	>61	6.11	0.81		
BAR	<20	5.42	1.33	0.532	0.752
	21–30	5.40	1.09		
	31–40	5.55	0.85		
	41–50	5.44	0.90		
	51–60	5.60	0.96		
	>61	5.61	1.02		
CUE	<20	6.03	0.74	1.900	0.093
	21–30	5.58	0.77		
	31–40	5.53	0.68		
	41–50	5.55	0.67		
	51–60	5.51	0.69		
	>61	5.73	0.76		
SEFF	<20	6.20	0.67	2.783	0.017 *
	21–30	5.64	0.73		
	31–40	5.60	0.65		
	41–50	5.67	0.63		
	51–60	5.54	0.64		
	>61	5.78	0.79		
WRV	<20	5.97	0.85	1.461	0.202
	21–30	5.77	0.87		
	31–40	5.69	0.73		
	41–50	5.66	0.74		
	51–60	5.71	0.94		
	>61	5.99	0.82		
PWOM	<20	6.09	0.97	1.231	0.294
	21–30	5.76	0.83		
	31–40	5.71	0.76		
	41–50	5.61	0.79		
	51–60	5.61	0.68		
	>61	5.74	0.891		
NWOM	<20	5.42	1.39	1.566	0.168
	21–30	5.29	1.27		
	31–40	5.38	1.00		
	41–50	5.10	1.05		
	51–60	4.91	1.08		
	>61	5.15	1.23		

* *p* < 0.05.

**Table 6 vaccines-11-01027-t006:** One-Way ANOVA Table of Education Level Variable.

Variables	Education	Mean	Standard Deviation	F	*p*
SUS	Middle school and below	5.69	0.92	0.412	0.744
	High school	5.54	0.78		
	Associate or bachelor	5.56	0.81		
	Master and above	5.64	0.78		
SER	Middle school and below	6.04	0.75	2.223	0.085
	High school	5.58	0.78		
	Associate or bachelor	5.63	0.76		
	Master and above	5.75	0.76		
BEN	Middle school and below	5.96	0.92	0.299	0.826
	High school	5.84	0.85		
	Associate or bachelor	5.83	0.89		
	Master and above	5.76	0.95		
BAR	Middle school and below	5.58	1.06	0.754	0.520
	High school	5.63	0.82		
	Associate or bachelor	5.45	0.97		
	Master and above	5.47	1.05		
CUE	Middle school and below	6.05	0.69	3.059	0.028 *
	High school	5.47	0.76		
	Associate or bachelor	5.59	0.69		
	Master and above	5.59	0.72		
SEFF	Middle school and below	6.13	0.71	4.528	0.004 *
	High school	5.55	0.71		
	Associate or bachelor	5.62	0.67		
	Master and above	5.78	0.67		
WRV	Middle school and below	5.80	1.00	0.525	0.666
	High school	5.69	0.72		
	Associate or bachelor	5.73	0.83		
	Master and above	5.82	0.76		
PWOM	Middle school and below	6.00	1.08	4.455	0.004 *
	High school	5.44	0.74		
	Associate or bachelor	5.77	0.77		
	Master and above	5.74	0.82		
NWOM	Middle school and below	5.28	1.43	0.391	0.759
	High school	5.21	1.00		
	Associate or bachelor	5.27	1.09		
	Master and above	5.13	1.27		

* *p* < 0.05.

**Table 7 vaccines-11-01027-t007:** One-Way ANOVA Table of Occupation Variable.

Variables	Occupation	Mean	Standard Deviation	F	*p*
SUS	Students	5.48	1.12	0.529	0.835
	Civil servant	5.60	0.79		
	Business	5.53	0.67		
	Workers	5.65	0.85		
	Manufacturing	5.56	0.70		
	Agriculture, forestry, fishery and animal husbandry	5.96	0.99		
	Service industry	5.64	0.80		
	Freelancer	5.45	0.89		
	Others	5.59	0.86		
SER	Students	5.83	0.77	1.179	0.310
	Civil servant	5.78	0.75		
	Business	5.66	0.68		
	Workers	5.82	0.83		
	Manufacturing	5.64	0.76		
	Agriculture, forestry, fishery and animal husbandry	6.23	0.75		
	Service industry	5.64	0.81		
	Freelancer	5.50	0.76		
	Others	5.56	0.78		
BEN	Students	5.97	1.04	1.276	0.254
	Civil servant	5.99	0.87		
	Business	5.95	0.92		
	Workers	5.91	0.79		
	Manufacturing	5.70	1.02		
	Agriculture, forestry, fishery and animal husbandry	6.19	1.00		
	Service industry	5.84	0.83		
	Freelancer	5.57	0.72		
	Others	5.69	0.94		
BAR	Students	5.25	1.36	2.003	0.045 *
	Civil servant	5.74	0.87		
	Business	5.66	0.82		
	Workers	5.71	0.77		
	Manufacturing	5.59	0.81		
	Agriculture, forestry, fishery and animal husbandry	6.07	1.02		
	Service industry	5.32	1.13		
	Freelancer	5.46	0.82		
	Others	5.31	0.91		
CUE	Students	5.69	0.94	0.751	0.647
	Civil servant	5.61	0.72		
	Business	5.55	0.68		
	Workers	5.51	0.71		
	Manufacturing	5.54	0.69		
	Agriculture, forestry, fishery and animal husbandry	6.06	0.95		
	Service industry	5.62	0.72		
	Freelancer	5.46	0.62		
	Others	5.62	0.74		
SEFF	Students	5.87	0.78	1.329	0.227
	Civil servant	5.77	0.80		
	Business	5.63	0.57		
	Workers	5.50	0.67		
	Manufacturing	5.66	0.68		
	Agriculture, forestry, fishery and animal husbandry	6.25	0.82		
	Service industry	5.66	0.68		
	Freelancer	5.57	0.68		
	Others	5.61	0.71		
WRV	Students	5.88	0.84	0.936	0.486
	Civil servant	5.85	0.77		
	Business	5.77	0.90		
	Workers	5.47	0.69		
	Manufacturing	5.73	0.74		
	Agriculture, forestry, fishery and animal husbandry	6.21	0.87		
	Service industry	5.77	0.83		
	Freelancer	5.66	0.76		
	Others	5.68	0.75		
PWOM	Students	6.02	0.94	2.139	0.031 *
	Civil servant	5.86	0.84		
	Business	5.83	0.71		
	Workers	5.53	0.87		
	Manufacturing	5.57	0.76		
	Agriculture, forestry, fishery and animal husbandry	6.18	0.79		
	Service industry	5.75	0.77		
	Freelancer	5.55	0.72		
	Others	5.51	0.92		
NWOM	Students	4.90	1.67	1.082	0.374
	Civil servant	5.47	1.04		
	Business	5.26	1.13		
	Workers	5.28	0.97		
	Manufacturing	5.38	0.90		
	Agriculture, forestry, fishery and animal husbandry	5.82	1.21		
	Service industry	5.10	1.25		
	Freelancer	5.22	0.84		
	Others	5.14	1.13		

* *p* < 0.05.

**Table 8 vaccines-11-01027-t008:** Regression Analysis of Independent Variables on Dependent Variables.

Regression Analysis	ß	R^2^	Adj R^2^	*t*	F	*p*
SUS→WRV	0.147	0.022	0.020	3.453	11.924	0.001 **
SER→WRV	0.199	0.040	0.038	4.705	22.139	0.000 ***
BEN→WRV	0.337	0.114	0.112	8.307	69.013	0.000 ***
BAR→WRV	0.396	0.157	0.155	10.010	100.202	0.000 ***
CUE→WRV	0.328	0.108	0.106	8.059	64.949	0.000 ***
SEFF→WRV	0.303	0.092	0.090	7.383	54.504	0.000 ***
PWOM→WRV	0.410	0.168	0.167	10.439	108.980	0.000 ***
NWOM→WRV	0.222	0.049	0.048	5.286	27.939	0.000 ***

** *p* < 0.01; *** *p* < 0.001.

**Table 9 vaccines-11-01027-t009:** The Moderate Effect of PWOM between Perceived Susceptibility and Willingness to Receive COVID-19 Vaccine.

Hierarchical Regression	Dependent Variable: Willingness to Receive the COVID-19 Vaccine					
	Model 1		Model 2		Model 3	
	ß	*t*	ß	*t*	ß	*t*
SUS	0.191	4.038	0.047	1.161	0.044	1.082
PWOM	-	-	0.578	14.401	0.577	14.324
SUS × NWOM	-		-		0.020	0.499
F	16.302		115.755		77.119	
R^2^	0.036		0.350		0.350	
ΔR^2^	0.036		0.314		0.000	
Result: PWOM has no moderate effect between Perceived Susceptibility and Willingness to Receive COVID-19 Vaccine. (*p* = 0.618 > 0.05)						

**Table 10 vaccines-11-01027-t010:** The Moderate Effect of PWOM between Perceived Seriousness and Willingness to Receive COVID-19 Vaccine.

Hierarchical Regression	Dependent Variable: Willingness to Receive the COVID-19 Vaccine					
	Model 1		Model 2		Model 3	
	ß	*t*	ß	*t*	ß	*t*
SER	0.271	0.5838	0.058	1.374	0.058	1.390
PWOM	-	-	0.568	13.557	0.566	13.432
SER × NWOM	-		-		0.021	0.538
F	34.086		116.170		77.415	
R^2^	0.073		0.351		0.351	
ΔR^2^	0.073		0.277		0.000	
Result: PWOM has no moderate effect between the Perceived Seriousness and Willingness to Receive COVID-19 Vaccine. (*p* = 0.591 > 0.05)						

**Table 11 vaccines-11-01027-t011:** The Moderate Effect of PWOM between Perceived Benefits of Taking Action and Willingness to Receive COVID-19 Vaccine.

Hierarchical Regression	Dependent Variable: Willingness to Receive the COVID-19 Vaccine					
	Model 1		Model 2		Model 3	
	ß	*t*	ß	*t*	ß	*t*
BEN	0.626	16.686	0.465	12.869	0.441	12.009
PWOM	-	-	0.404	11.178	0.410	11.439
BEN × NWOM	-		-		−0.098	−2.901
F	278.407		241.709		166.722	
R^2^	0.392		0.529		0.538	
ΔR^2^	0.392		0.137		0.009	
Result: PWOM has a moderate effect between the Perceived Benefits of Taking Action and Willingness to Receive COVID-19 Vaccine. (*p* = 0.004 < 0.05)						

**Table 12 vaccines-11-01027-t012:** The Moderate Effect of PWOM between Perceived Barriers of Taking Action and and Willingness to Receive COVID-19 Vaccine.

Hierarchical Regression	Dependent Variable: Willingness to Receive the COVID-19 Vaccine					
	Model 1		Model 2		Model 3	
	ß	*t*	ß	*t*	ß	*t*
BAR	0.352	7.796	0.265	7.096	0.286	7.471
PWOM	-	-	0.548	14.684	0.545	14.659
BAR × NWOM	-		-		−0.086	−2.273
F	60.784		153.335		104.937	
R^2^	0.124		0.416		0.423	
ΔR^2^	0.124		0.293		0.007	
Result: PWOM has no moderate effect between Perceived Barriers to Taking Action and Willingness to Receive COVID-19 Vaccine. (*p* = 0.023 < 0.05)						

**Table 13 vaccines-11-01027-t013:** The Moderate Effect of PWOM between Cues to Action and Willingness to Receive COVID-19 Vaccine.

Hierarchical Regression	Dependent Variable: Willingness to Receive the COVID-19 Vaccine					
	Model 1		Model 2		Model 3	
	ß	*t*	ß	*t*	ß	*t*
CUE	0.494	11.789	0.238	5.204	0.240	5.246
PWOM	-	-	0.457	10.003	0.458	10.020
CUE × NWOM	-		-		−0.030	−0.776
F	138.990		135.492		90.445	
R^2^	0.244		0.387		0.387	
ΔR^2^	0.244		0.143		0.001	
Result: PWOM has no moderate effect between Cues to Action and Willingness to Receive COVID-19 Vaccine. (*p* = 0.438 > 0.05)						

**Table 14 vaccines-11-01027-t014:** The Moderate Effect of PWOM between Self-Efficacy and Willingness to Receive COVID-19 Vaccine.

Hierarchical Regression	Dependent Variable: Willingness to Receive the COVID-19 Vaccine					
	Model 1		Model 2		Model 3	
	ß	*t*	ß	*t*	ß	*t*
SEFF	0.436	10.058	0.203	4.700	0.202	4.687
PWOM	-	-	0.494	11.468	0.494	11.437
SEFF × NWOM	-		-		−0.001	−0.032
F	101.166		131.662		87.571	
R^2^	0.190		0.380		0.380	
ΔR^2^	0.190		0.190		0.000	
Result: PWOM has no moderate effect between Self-Efficacy and Willingness to Receive COVID-19 Vaccine. (*p* = 0.975 > 0.05)						

**Table 15 vaccines-11-01027-t015:** The Moderate Effect of NWOM between Perceived Susceptibility and Willingness to Receive COVID-19 Vaccine.

Hierarchical Regression	Dependent Variable: Willingness to Receive the COVID-19 Vaccine					
	Model 1		Model 2		Model 3	
	ß	*t*	ß	*t*	ß	*t*
SUS	0.191	4.038	0.163	3.415	0.172	3.582
NWOM	-	-	0.143	3.004	0.141	2.946
SUS × NWOM	-		-		0.074	1.581
F	16.302		12.813		9.405	
R^2^	0.036		0.056		0.062	
ΔR^2^	0.036		0.020		0.005	
Result: NWOM has no moderate effect between Perceived Susceptibility and Willingness to Receive COVID-19 Vaccine. (*p* = 0.115 > 0.05)						

**Table 16 vaccines-11-01027-t016:** The Moderate Effect of NWOM between Perceived Seriousness and Willingness to Receive COVID-19 Vaccine.

Hierarchical Regression	Dependent Variable: Willingness to Receive the COVID-19 Vaccine					
	Model 1		Model 2		Model 3	
	ß	*t*	ß	*t*	ß	*t*
SER	0.271	5.838	0.244	5.185	0.249	5.175
NWOM	-	-	0.123	2.605	0.117	2.379
SER × NWOM	-		-		0.023	0.474
F	34.086		20.664		13.826	
R^2^	0.073		0.088		0.088	
ΔR^2^	0.073		0.014		0.000	
Result: NWOM has no moderate effect between Perceived Seriousness and Willingness to Receive COVID-19 Vaccine. (*p* = 0.636 > 0.05)						

**Table 17 vaccines-11-01027-t017:** The Moderate Effect of NWOM between Perceived Benefits of Taking Action and and Willingness to Receive COVID-19 Vaccine.

Hierarchical Regression	Dependent Variable: Willingness to Receive the COVID-19 Vaccine					
	Model 1		Model 2		Model 3	
	ß	*t*	ß	*t*	ß	*t*
BEN	0.626	16.686	0.614	16.244	0.610	16.041
NWOM	-	-	0.084	2.210	0.080	2.104
BEN × NWOM	-		-		0.031	0.820
F	278.407		142.901		95.419	
R^2^	0.392		0.399		0.400	
ΔR^2^	0.392		0.007		0.001	
Result: NWOM has a moderate effect between Perceived Benefits of Taking Action and Willingness to Receive COVID-19 Vaccine. (*p* = 0.413 > 0.05)						

**Table 18 vaccines-11-01027-t018:** The Moderate Effect of NWOM between Perceived Barriers of Taking Action and Willingness to Receive COVID-19 Vaccine.

Hierarchical Regression	Dependent Variable: Willingness to Receive the COVID-19 Vaccine					
	Model 1		Model 2		Model 3	
	ß	*t*	ß	*t*	ß	*t*
BAR	0.352	7.796	0.340	6.778	0.403	8.294
NWOM	-	-	0.027	0.544	0.088	1.815
BAR × NWOM	-		-		0.311	6.832
F	60.784		30.490		38.044	
R^2^	0.124		0.124		0.210	
ΔR^2^	0.124		0.001		0.086	
Result: NWOM has a moderate effect between Perceived Barriers to Taking Action and Willingness to Receive COVID-19 Vaccine. (*p* = 0.000 < 0.05)						

**Table 19 vaccines-11-01027-t019:** The Moderate Effect of NWOM between Cues to Action and Willingness to Receive COVID-19 Vaccine.

Hierarchical Regression	Dependent Variable: Willingness to Receive the COVID-19 Vaccine					
	Model 1		Model 2		Model 3	
	ß	*t*	ß	*t*	ß	*t*
CUE	0.494	11.789	0.479	11.149	0.476	11.057
NWOM	-	-	0.066	1.543	0.084	1.734
CUE × NWOM	-		-		−0.037	−0.793
F	138.990		70.907		47.440	
R^2^	0.244		0.248		0.249	
ΔR^2^	0.244		0.004		0.001	
Result: NWOM has no moderate effect between Cues to Action and Willingness to Receive COVID-19 Vaccine. (*p* = 0.428 > 0.05)						

**Table 20 vaccines-11-01027-t020:** The Moderate Effect of NWOM between Self-Efficacy and Willingness to Receive COVID-19 Vaccine.

Hierarchical Regression	Dependent Variable: Willingness to Receive the COVID-19 Vaccine					
	Model 1		Model 2		Model 3	
	ß	*t*	ß	*t*	ß	*t*
SEFF	0.436	10.058	0.420	9.299	0.419	9.213
NWOM	-	-	0.056	1.230	0.059	1.200
SEFF × NWOM	-		-		−0.008	−0.171
F	101.166		51.400		34.199	
R^2^	0.190		0.193		0.193	
ΔR^2^	0.190		0.003		0.000	
Result: NWOM has no moderate effect between Self-Efficacy and Willingness to Receive COVID-19 Vaccine. (*p* = 0.864 > 0.05)						

## Data Availability

The datasets generated during the current study are not publicly available but are available from the corresponding author upon reasonable request.

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
