# Peer review of "Exploring the Moderating Effect of Positive and Negative Word-of-Mouth on the Relationship between Health Belief Model and the Willingness to Receive COVID-19 Vaccine"

_vaccines, 2023, doi:10.3390/vaccines11061027_

Round 1

Reviewer 1 Report

The study examines the impact of word-of-mouth on vaccine uptake by utilizing a health belief model and a modified word-of-mouth scale. However, the paper is overly long and poorly written, making it unsuitable for publication in Vaccines without significant revisions. The authors should aim to condense the paper to one-quarter of its current length and improve its clarity by addressing the missing citations and rephrasing it to align with the style of a public health journal. At present, the paper reads more like a business or economics article, which is not suitable for Vaccines. To enhance the quality of their work, the authors should consider the feedback provided and make necessary changes before submitting it to any journal.

1) Introduction is very poorly written. The paper spends 4 pages in the introduction, going over the historical background of the pandemic, etc. This is absolutely not necessary! Authors can start on line 100 and take it from there, to clearly and succinctly introduce HBM, and present a research hypothesis.

2)  The paper spends EIGHT pages in the literature review! This is absurd.

3) The paper lacks clear organization and clear hypothesis to test. Instead, the paper comes up with 7 main hypotheses and 13 additional sub-hypothesis! The paper should focus on 1-2 main hypotheses, and discuss the rest in the results.

4) The 9 different scales in the health belief model are interesting, but conceptually and methodologically I am not convinced they bring sufficient value to vaccine uptake, despite what the authors mention in the paper. These tests are highly correlated and not surprising the results on table 11 just do a linear regression for each of the variables (independent from each other).

5) The conclusion could be used to describe the main results and how they contract with the related literature.

6) Very much all of section 4.6 is just a bunch of tables without a single explanation.

7) Significant lack of references on vaccine uptake is missing. This includes work done on trust and consumption of information and vaccine uptake, many of them published already on vaccines. Relationships with those sources and comparison, at last in the conclusion is necessary.

Overall, this is a paper that does not fit vaccines well. Once the authors are able to cut the paper and have a clear hypothesis significantly, they could reconsider at a different public health journal.

All the introductions have spacing issues, including in the parenthesis, for instance L92 patients( should be patients (. Lines 106, 114, ... have the same issue

t test, t-test, T-test are all written inconsistently, and often next to each other (e.g., L726-732)

The paper is written in decent English, but some minor expressions here and there are unusual, one of these examples include "Vaccine Injection Willingness" which may be called vaccine uptake. I recommend they send this to a native English speaker for review

Author Response

Dear reviewer,

Thank you for your comments and suggestions.

The revised manuscript, which has uploaded.

Regarding your comments, we have replied point by point.

Please, review the contents in the attached file.

Response to Reviewer 1 Comments

Point 1) Introduction is very poorly written. The paper spends 4 pages in the introduction, going over the historical background of the pandemic, etc. This is absolutely not necessary! Authors can start on line 100 and take it from there, to clearly and succinctly introduce HBM, and present a research hypothesis.

Response: In this paper, unnecessary content and introductions have been removed from the Introduction section to concisely present the Health Belief Model (HBM) and Word-of-Mouth (WOM). Moreover, the numerous research hypotheses have been restructured.

Point 2) The paper spends EIGHT pages in the literature review! This is absurd.

Response: The literature review has been thoroughly revised in the original version, providing a brief introduction to the variables in this study.

Point 3) The paper lacks clear organization and clear hypothesis to test. Instead, the paper comes up with 7 main hypotheses and 13 additional sub-hypothesis! The paper should focus on 1-2 main hypotheses, and discuss the rest in the results.

Response: We have proposed and discussed three clearer hypotheses (H1-H3), each of which has been verified.

Point 4) The 9 different scales in the health belief model are interesting, but conceptually and methodologically I am not convinced they bring sufficient value to vaccine uptake, despite what the authors mention in the paper. These tests are highly correlated and not surprising the results on table 11 just do a linear regression for each of the variables (independent from each other).

Response: This study aims to highlight the statistical performance of the variables in the Health Belief Model in relation to positive and negative word-of-mouth. If approved by the reviewers, we hope to expand and improve upon this in future research.

Point 5) The conclusion could be used to describe the main results and how they contract with the related literature.

Response: In the conclusion, we have compared our results with those of recent literature.

Point 6) Very much all of section 4.6 is just a bunch of tables without a single explanation.

Response: All statistical analysis results for the tables have been explained, and we have indicated whether they support or reject hypotheses in this research.

Point 7) Significant lack of references on vaccine uptake is missing. This includes work done on trust and consumption of information and vaccine uptake, many of them published already on vaccines. Relationships with those sources and comparison, at last in the conclusion is necessary.

Response: Recent literature has been included in the statement for comparison.

NOTE:The paragraph spacing and statistical t-test formatting have been unified, thank you for the correction.

In the end, we hope it meets your expectations,

and we appreciate the time you have taken to review this work.

Pei-yun Chiang

Reviewer 2 Report

This paper discusses a very significant issue in the present world about the willingness of the persons to receive the vaccination and the associated factors influencing this decision of theirs. The whole manuscript has been written really well and in comprehensible English language easy to understand for all the readers. It covers all the necessary aspects of the topic and is extensively written. The authors have discussed their findings and have correlated it with other good studies. 

Author Response

Dear Reviewer,

Thank you for your recognition of this research and for providing valuable comments and suggestions.

The revised manuscript, which has undergone a thorough editing process and improved English writing quality, which has uploaded.

Unnecessary paragraphs have been removed without losing the original meaning.

In addition, the non-significant demographic analysis has been explained in the conclusion.

Finally, thank you for your reminder; we have defined the abbreviations in the abstract.

We hope it meets your expectations, and we appreciate the time you have taken to review this work.

Pei-yun Chiang

Reviewer 3 Report

This paper has an excellent premise and excellent study design. The primary issue is presentation for this journal.  This is written like a master's thesis, which is fine, except, as a reader, I don't want all of the background. The intro needs to be 5 paragraphs max (so cut it down by well over half).   Next, if the results are not significant (e.g. marital status), present as supplemental.  Lastly the authors set the stage for guidelines based on this study and then don't present any. Either remove that expectation or give some public health measures.  Overall this study is excellent but the presentation is poor.  The story gets lost in the detail. The intro needs to be 1/3 the length with references to get more detail. The results need to be 1/2 the length with supplemental information where it doesn't facilitate the story.  The conclusion needs to be expanded to include recommendations.  This needs MAJOR rework with the goal of telling the reader the key points of a cohesive study, which the authors have done.

English language skills are acceptable with only minor points like defining your abbreviations before you start using them (e.g. in the abstract HBM is used and in the intro WOM is not defined).

Author Response

Dear Reviewer,

Thank you for your recognition of this research and for providing valuable comments and suggestions.

The revised manuscript, which has undergone a thorough editing process and improved English writing quality, , which has uploaded.

Unnecessary paragraphs have been removed without losing the original meaning.

In addition, the non-significant demographic analysis has been explained in the conclusion.

Finally, thank you for your reminder; we have defined the abbreviations in the abstract.

We hope it meets your expectations, and we appreciate the time you have taken to review this work.

Pei-yun Chiang

Reviewer 4 Report

Thank you for such important research, which was well done with such rigorous methodology. 

The comments I have are very minor and just grammatically related. I recommend extensive proofreading to improve the quality of the paper. 

Author Response

Dear Reviewer,

Thank you for your comments and suggestions.

The revised manuscript, which has uploaded.

We hope it meets your expectations,

and we appreciate the time you have taken to review this work.

Pei-yun Chiang

Round 2

Reviewer 1 Report

This interesting paper evaluates vaccine uptake using a health belief model in Taiwan. I am glad to see a significantly revised version where finally the authors cut in half the useless material of the previous version. This makes for a better understanding of the paper, whereby word of mouth is shown to be associated with vaccine uptake. The health belief model is interesting and well-accepted, so I am glad to see it in practice. Happy to recommend acceptance.